# Unusual Dependence of the Diamond Growth Rate on the Methane Concentration in the Hot Filament Chemical Vapor Deposition Process

**DOI:** 10.3390/ma14020426

**Published:** 2021-01-16

**Authors:** Byeong-Kwan Song, Hwan-Young Kim, Kun-Su Kim, Jeong-Woo Yang, Nong-Moon Hwang

**Affiliations:** 1Department of Materials Science and Engineering, Seoul National University, 1 Gwanak-ro, Gwanak-gu, Seoul 08826, Korea; sbk32131832@snu.ac.kr (B.-K.S.); welcome777@snu.ac.kr (H.-Y.K.); lstatsl@snu.ac.kr (K.-S.K.); jwoo5432@snu.ac.kr (J.-W.Y.); 2Research Institute of Advanced Materials, 599 Gwanak-ro, Gwanak-gu, Seoul 08826, Korea

**Keywords:** unusual deposition behavior, diamond deposition, hot-filament chemical vapor deposition (CVD), graphite coating on filament, electron emission

## Abstract

Although the growth rate of diamond increased with increasing methane concentration at the filament temperature of 2100 °C during a hot filament chemical vapor deposition (HFCVD), it decreased with increasing methane concentration from 1% CH_4_ –99% H_2_ to 3% CH_4_ –97% H_2_ at 1900 °C. We investigated this unusual dependence of the growth rate on the methane concentration, which might give insight into the growth mechanism of a diamond. One possibility would be that the high methane concentration increases the non-diamond phase, which is then etched faster by atomic hydrogen, resulting in a decrease in the growth rate with increasing methane concentration. At 3% CH_4_ –97% H_2_, the graphite was coated on the hot filament both at 1900 °C and 2100 °C. The graphite coating on the filament decreased the number of electrons emitted from the hot filament. The electron emission at 3% CH_4_ –97% H_2_ was 13 times less than that at 1% CH_4_ –99% H_2_ at the filament temperature of 1900 °C. The lower number of electrons at 3% CH_4_ –97% H_2_ was attributed to the formation of the non-diamond phase, which etched faster than diamond, resulting in a lower growth rate.

## 1. Introduction

The low pressure synthesis of diamond has been studied extensively and intensively with various types of chemical vapor deposition (CVD) using hot filament or plasma [1,2,3], since the deposition of diamond in the form of thin films has opened a variety of new applications, such as cutting tools, sensors, heat spreaders, and optical windows [4,5,6]. Although the CVD diamond films are produced commercially, some deposition behaviors have not been understood clearly. One popular explanation is that atomic hydrogen plays a main role for growing diamond. That is, atomic hydrogen, generated by various sources, such as hot filament or plasma discharge, continuously makes reactive sites on the diamond surface for carbon absorption into the diamond structure [7,8,9,10,11]. However, one phenomenon in the CVD diamond process is hard to explain solely by atomic hydrogen hypothesis, such as simultaneous diamond deposition and graphite etching [12,13]. The process using graphite etching as a carbon source for diamond deposition has also been studied [14]. However, graphite is more stable in the low-pressure process than diamond, and comparing the chemical potential of carbon in gas, diamond and graphite indicates that, if stable graphite etches away into the gas phase, less stable diamond should also etch away into the gas phase. This puzzling phenomenon is often called the “thermodynamic paradox” [15,16]. Therefore, the explanation by atomic hydrogen has a critical weak point because it would violate the second law of thermodynamics. To avoid the thermodynamic paradox of simultaneous diamond deposition and graphite etching, Hwang et al. suggested a theory of charged nanoparticles [17,18,19,20].

In this paper, we introduce the unusual deposition behavior of a diamond, which is difficult to explain by the conventional concept of crystal growth. In the CVD process, the growth rate of films generally increases with increasing concentration of precursor. In the hot filament CVD (HFCVD) process of diamond, the growth rate of diamond is proportional to the amount of methane gas introduced into the chamber. We observed this deposition behavior at the filament temperature of 2100 °C. However, at the filament temperature of 1900 °C, we observed that the deposition rate of diamond decreased with increasing concentration of methane from 1% CH_4_ –99% H_2_ to 3% CH_4_ –97% H_2_. Since the phenomenon is unusual, we checked and found that it is certainly reproducible. The purpose of this paper is to understand this unusual dependence of the deposition rate on methane concentration.

To understand this unusual phenomenon, we examined how the deposition environment changes at methane concentrations of 1% CH_4_ –99% H_2_ and 3% CH_4_ –97% H_2_ at filament temperatures of 1900 °C and 2100 °C. We found out that this unusual deposition behavior is related to the filament coated by the graphite. However, the unusual deposition behavior, which could not be explained by filament coating alone, turned out to be more directly related to the number of electrons emitted from the hot filament, which could be measured as electric current by the electrometer.

## 2. Materials and Methods

A hot filament CVD reactor was used for diamond deposition. The filament consisted of three tungsten wires of ø 0.5 mm, twisted into a nine-turn coil of ø 8 mm. The reactor pressure was 20 Torr, and the filament temperature varied from 1900 °C to 2100 °C depending on the experimental purpose. CH_4_ and H_2_ were supplied as a gas mixture at 1 standard cubic centimeter per minute (sccm) and 99 sccm, or at 3 sccm and 97 sccm, respectively, using a mass flow controller.

A silicon wafer of 1 cm × 1 cm was used as a substrate. By adjusting the distance between the hot filament and substrate, the substrate temperature could be targeted at 900 °C. In order to deposit isolated diamond particles, the silicon substrate was not pretreated to prevent the film deposition. The microstructure of the deposited diamond was observed by field-emission scanning electron microscopy (FE-SEM; SU70; Hitachi Ltd., Tokyo, Japan), with an acceleration voltage of 15 kV.

To capture the diamond nanoparticles generated in the gas phase, we installed a capturing apparatus in the reactor, which was reported by Park et al. [21]. Figure 1 shows a schematic of the capturing apparatus. The SiO membrane on the Cu grid for the transmission electron microscope (TEM) (SiO Type-A, Ted Pella, Inc., Redding, CA, USA), on which nanoparticles were captured for 15 s, was placed in the hole at the tip of the quartz probe. The probe could be moved from the sidewall of the chamber to the capture position at 550–600 °C and then moved away to the sidewall using the feedthrough. The SiO membrane tended to be damaged during HFCVD, and the capturing temperature of 550–600 °C was the maximum to allow the capturing time of 15 s without severe damage to the membrane. The capturing temperature was measured by a K-type thermocouple.

We measured the electric current of thermionic electrons emitted from the hot filament with an iron probe installed in the feedthrough, connected to a picoammeter (model 6487; Keithley Instruments, Cleveland, OH, USA) outside the chamber.

## 3. Results

### 3.1. Unusual Dependence on Diamond Growth Rate

Figure 2 shows the diamond particles deposited on the untreated silicon substrate at the filament temperature of 2100 °C with the gas mixtures of 1% CH_4_ –99% H_2_ and 3% CH_4_ –97% H_2_. Figure 2a,b shows the low magnification FE-SEM images of diamond particles deposited for 4 h and 8 h, respectively, at 1% CH_4_ –99% H_2_. Figure 2c,d shows the high magnification FE-SEM images, respectively, of Figure 2a,b. Figure 2e,f shows the low magnification FE-SEM images of diamond particles deposited for 4 h and 8 h, respectively, at 3% CH_4_ –97% H_2_. Figure 2g,h shows the high magnification FE-SEM images, respectively, of Figure 2e,f. The number densities of diamond particles in Figure 2a,b are, respectively, 53/mm^2^ and 145/mm^2^, while those of Figure 2e,f are 314/mm^2^ and 560/mm^2^, respectively. The average sizes of diamond particles in Figure 2a,b are, respectively, ~3.1 μm and ~5.7 μm, while those of Figure 2e,f are ~4.8 μm and ~8.9 μm, respectively. With increasing methane concentration from 1 to 3%, the number density and the size of diamond particles increased, as expected.

On the other hand, the dependence of the number density and the size of diamond particles on methane concentration is reversed at the filament temperature of 1900 °C, as shown in Figure 3, where the diamond particles were deposited at the filament temperature of 1900 °C with the gas mixtures of 1% CH_4_ –99% H_2_ and 3% CH_4_ –97% H_2_. Figure 3a,b shows the low magnification FE-SEM images of diamond particles deposited for 4 h and 8 h, respectively, at 1% CH_4_ –99% H_2_, with Figure 3c,d being the respective high magnification FE-SEM images. Figure 3e,f shows the low magnification FE-SEM images of diamond particles deposited for 4 h and 8 h, respectively, at 3% CH_4_ –97% H_2_, with Figure 3g,h being the respective high magnification FE-SEM images. The number densities of diamond particles in Figure 3a,b are, respectively, 107/mm^2^ and 161/mm^2^, while those of Figure 3e,f are 115/mm^2^ and 138/mm^2^, respectively. The average sizes of diamond particles in Figure 3a,b are, respectively, ~2.2 μm and ~4.3 μm, while those of Figure 3e,f are ~1.5 μm and ~3.0 μm, respectively. With increasing methane concentration from 1 to 3%, the number density and the average size of diamond particles did not increase but decreased, which is contrary to our expectation. These results were reproducible. Additional trends of the observed diamond particles were summarized in Appendix A.

Therefore, the dependence of the growth rate on the methane concentration is reversed between the filament temperatures of 2100 °C and 1900 °C. Normally, the flux for the deposition would increase with increasing methane concentration. Why does the deposition rate decrease with increasing methane concentration at a filament temperature of 1900 °C? One possible explanation for Figure 3, where the deposition rate of diamond decreases with increasing methane concentration, would be that the etching rate in the condition using 3% CH_4_ –97% H_2_ is higher than that using 1% CH_4_ –99% H_2_. The question then arises as to why the etching rate in the condition using 3% CH_4_ –97% H_2_ is higher than that using 1% CH_4_ –99% H_2_. As an answer to this question, we can assume that if the diamond deposited in the condition using 3% CH_4_ –97% H_2_ had a higher content of a non-diamond phase, the diamond deposited in the condition using 3% CH_4_ –97% H_2_ would be etched faster than that using 1% CH_4_ –99% H_2_. To examine the crystallinity of deposited particles shown in Figure 2 and Figure 3, the local area of each particle was analyzed by a micro Raman spectrometer with a spot size of 1 μm. Appendix A shows the Raman spectra of each particle deposited under the filament temperature of 2100 °C and 1900 °C at the gas mixture of 1% CH_4_ –99% H_2_ or 3% CH_4_ –97% H_2_.

### 3.2. Surface of the Filament

Why does this unusual deposition behavior occur at the filament temperature of 1900 °C, and why doesn’t it happen at 2100 °C? The reason might be the difference in the quality of deposited diamond between the filament temperatures of 1900 °C and 2100 °C. In relation to this possibility, Sommer et al. [22] suggested that, under the condition where the hot filament is coated by a graphite, non-diamond is formed. To check this possibility, we tried to confirm whether the filament after deposition was coated by a graphite or not. When the filament temperature was 2100 °C, the filament was not coated with the graphite at 1% CH_4_ –99% H_2_, as shown in Figure 4a, but it was almost fully coated with the graphite at 3% CH_4_ –97% H_2_, as shown in Figure 4b. When the filament temperature was 1900 °C, the filament was only partially coated with the graphite at 1% CH_4_ –99% H_2_, as shown in Figure 4c, but it was almost fully coated with the graphite at 3% CH_4_ –97% H_2_, as shown in Figure 4d.

These coating behaviors of the filament agree with the thermodynamic calculations by a Thermo-Calc software (Royal Institute of Technology, Stockholm, Sweden) using a Scientifica Group Thermodata Europe (SGTE) database [23], as shown in Figure 5. Figure 5 shows that, at 3% CH_4_ –97% H_2_, the graphite would precipitate even at 2300 °C. Under this condition of carbon precipitation, the filament would be coated by the graphite.

If non-diamond is coated under the condition where the filament is coated by a graphite, as suggested by Sommer et al. [22], for the filament temperature of 2100 °C, the diamond deposited at 1% CH_4_ –99% H_2_ would not contain the non-diamond phase, whereas the diamond deposited at 3% CH_4_ –97% H_2_ would. Figure 2 shows that the diamond deposited at 1% CH_4_ –99% H_2_ had faceted surfaces, but the diamond deposited at 3% CH_4_ –97% H_2_ had a ball-like and cauliflower shape with numerous nanonodules on the surface. The ball-like diamond contains some non-diamond phase and its etching rate would be higher than that of the diamond with faceted surfaces. In spite of the high etching rate of the ball-like diamond, Figure 2 shows that the size of the ball-like diamond is larger than that of the diamond with faceted surfaces, indicating that the growth rate of the former is higher than that of the latter. The higher flux of the diamond deposited at 3% CH_4_ –97% H_2_ than that at 1% CH_4_ –99% H_2_ outweighs the higher etching rate of the ball-like diamond than that of the diamond with faceted surfaces.

However, for the filament temperature of 1900 °C, the diamond deposited at 1% CH_4_ –99% H_2_ had a ball-like shape that would contain some non-diamond phase and the diamond deposited at 3% CH_4_ –97% H_2_ also had a ball-like shape, as shown in Figure 3. The lower growth rate of the ball-like diamond deposited at 3% CH_4_ –97% H_2_ indicates that its etching rate is higher than that deposited at 1% CH_4_ –99% H_2_. This higher etching rate would be attributed to the higher content of the non-diamond phase in the ball-like diamond deposited at 3% CH_4_ –97% H_2_ than that deposited at 1% CH_4_ –99% H_2_.

Comparing Figure 2g,h with Figure 3g,h, the growth rate of the ball-like diamond deposited at 2100 °C was much higher than that deposited at 1900 °C at 3% CH_4_ –97% H_2_, indicating that the etching rate of the ball-like diamond deposited at 2100 °C is much lower than that deposited at 1900 °C. In other words, the ball-like diamond deposited at 1900 °C contains a larger amount of the non-diamond phase, which can be etched at a much higher rate. It should be noted that the amount of atomic hydrogen formed at the filament temperature of 2100 °C would be larger than that of 1900 °C, and thereby, the etching rate at 2100 °C is expected to be higher than that at 1900 °C.

It should be emphasized that the ball-like diamond in Figure 2g,h is larger than even those in Figure 3c,d, indicating that the growth rate of the former is higher than that of the latter or that the etching rate of the former is lower than that of the latter. Considering all these results, the etching rate of the ball-like diamond deposited under the condition of filament coating by the graphite can be drastically different. The etching rate depends sensitively on whether the filament temperature was 2100 °C or 1900 °C.

Here, the main role of the filament has been suggested to generate atomic hydrogen, which would etch the non-diamond phase [11]. As mentioned in the Section 1, graphite etching and diamond deposition occur simultaneously during the diamond synthesis [12,13]. It is well known that atomic hydrogen, generated after being dissociated with molecular hydrogen by hot filament or by plasma discharge, if plasma is used for the gas activation, is a main etchant for graphite [9,10,11]. The amount of atomic hydrogen generated at 2100 °C would be much larger than that at 1900 °C. Since the etching rate of the ball-like diamond deposited at 2100 °C was much less than that at 1900 °C, the role of the filament temperature cannot be explained by the generation of atomic hydrogen alone.

### 3.3. Electron Emission From the Hot-Filament

As a possible role of the filament temperature, we can consider the electrons thermally emitted from the hot filament. The number of emitted electrons at different conditions of diamond deposition can be measured by electric current. Figure 6 shows the current measured at filament temperatures of 1900 °C, 2000 °C, and 2100 °C at 1% CH_4_ –99% H_2_ and 3% CH_4_ –97% H_2_. Figure 6 shows that the negative current increases with increasing filament temperature and that it is larger at 1% CH_4_ –99% H_2_ than at 3% CH_4_ –97% H_2_. At 1900 °C, the current was −7.10 μA/cm^2^ at 1% CH_4_ –99% H_2_ and −0.49 μA/cm^2^ at 3% CH_4_ –97% H_2_. The negative current measured at 1% CH_4_ –99% H_2_ was 13 times larger than that at 3% CH_4_ –97% H_2_. At 2000 °C, the current was −9.39 μA/cm^2^ at 1% CH_4_ –99% H_2_ and −2.55 μA/cm^2^ at 3% CH_4_ –97% H_2_. Additionally, here, the negative current measured at 1% CH_4_ –99% H_2_ was 2.68 times larger than that at 3% CH_4_ –97% H_2_. Lastly, at 2100 °C, the current was −17.3 μA/cm^2^ at 1% CH_4_ –99% H_2_ and −5.57 μA/cm^2^ at 3% CH_4_ –97% H_2_. Here, the negative current measured at 1% CH_4_ –99% H_2_ was 2.1 times larger than that at 3% CH_4_ –97% H_2_.

At 1% CH_4_ –99% H_2_, the negative current of −7.10 μA/cm^2^ increased to −17.3 μA/cm^2^ when the filament temperature increased from 1900 °C to 2100 °C. At 3% CH_4_ –97% H_2_, the negative current of −0.49 μA/cm^2^ increased to −5.57 μA/cm^2^ when the filament temperature increased from 1900 °C to 2100 °C. The increase of the negative current with increasing filament temperature can be understood by considering the equation of thermionic emission by Richardson [24].

The dramatic decrease in the measured current when the methane concentration changed from 1% CH_4_ –99% H_2_ to 3% CH_4_ –97% H_2_ may have been caused by the coating of the filament surface by the graphite, as shown in Figure 4. The electron emission described by the Richardson–Dushman equation [24,25] depends on the work function of the surface. The filament surface not coated by the graphite would be tungsten carbide, which has a work function of 3.6 eV [26]. When the filament surface is coated by carbon, the surface would be the graphite, which has a work function of 4.6 eV [27]. Due to the different work functions, the negative current was smaller at 3% CH_4_ –97% H_2_ than at 1% CH_4_ –99% H_2_.

From the above results, we can see that the diamond quality was related to the current. For example, the highest etching rate of the ball-like diamond deposited at 1900 °C and the gas mixture of 3% CH_4_ –97% H_2_ seemed to match the smallest value of the negative current of −0.49 μA/cm^2^. Moreover, the lowest etching rate of the diamond with faceted surfaces deposited at 2100 °C and the gas mixture of 1% CH_4_ –99% H_2_ seemed to match the largest value of −17.3 μA/cm^2^.

Let’s consider how these negative charges can affect the HFCVD process. In the historic Wilson cloud chamber experiment, it is well established that ions induce nucleation [28,29,30]. Since the electrostatic energy of ions is expressed by e^2^/r, where e is the electronic charge and r is the radius of an ion, the attachment of supersaturated atoms to ions would decrease the electrostatic energy. Therefore, it would be a spontaneous process for charged carbon clusters to form in the HFCVD diamond process, considering that an enormous amount of negatively charged electrons or ions is generated from the hot filament. The nucleation barrier, which should be overcome by spontaneously formed charged clusters to become a critical nucleus, would be much smaller than that of the neutral nucleation. Therefore, the so-called ion-induced nucleation would occur in the HFCVD diamond process. Charged nuclei would then be produced in the gas phase.

The gas phase nucleation is also manifested indirectly by the etching of graphite or diamond during the HFCVD process. Considering the CVD phase diagram of the C–H system [16], the gas mixture of 1% CH_4_ –99% H_2_ and 3% CH_4_ –97% H_2_ falls into the two-phase region of gas + graphite, which means that the driving force is for the deposition of carbon, which can be graphite or diamond. In the diamond HFCVD process, it is well known that graphite is etched at the substrate temperature of ~900 °C. Considering that both the composition of the gas mixture and the substrate temperature are in the two phase region of the C–H phase diagram, graphite or diamond cannot be etched unless gas phase nucleation occurs. In other words, the driving force becomes etching or deposition, depending on whether gas phase nucleation occurs or not. The gas-phase nucleation was confirmed in the diamond CVD process by many scientists [31,32,33,34]. 

As to the correlation between the filament temperature and the diamond quality, there is a possibility that the number of electric charges might affect the stability of diamond nanoparticles formed in the gas phase. To confirm this possibility, we captured diamond nanoparticles in the gas phase at filament temperatures of 1900 °C and 2100 °C at the gas mixture of 3% CH_4_ –97% H_2_. 

### 3.4. Comparing the Captured Nanoparticles and the Diamond Particles

Figure 7 shows TEM images of carbon nanoparticles captured for 15 s on the SiO membrane of the TEM grid, which were placed 30 mm below the hot filament. Figure 7a,b shows the scanning TEM (STEM) images of nanoparticles captured at 1900 °C and 2100 °C, respectively. The STEM images shown in Figure 7 had a much higher contrast between the nanoparticles and the membrane by the STEM mode than by the high resolution TEM (HRTEM) mode when smaller nanoparticles were analyzed [35]. The white spots shown in Figure 7a,b are nanoparticles; the crystalline nanoparticles tend to appear bright because of incoherently scattered electrons [36]. The number densities of nanoparticles captured at 1900 °C and 2100 °C were 58 per μm^2^ and 26 per μm^2^, respectively. Figure 7c,d shows high resolution TEM (HRTEM) images of nanoparticles captured at 1900 °C and 2100 °C, respectively, with the inset of a fast Fourier transformation (FFT) image.

Nanoparticles, which were captured at 1900 °C, were polycrystalline with the size of ~12 nm. D-spacings of these polycrystalline nanoparticles consisted of 2.06 Å, 2.2 Å, and 2.5 Å, which were measured within an error of 5% by FFT of the HRTEM images. From a measurement of more than 25 nanoparticles, the observed relative fractions of 2.06 Å, 2.2 Å, and 2.5 Å were 13%, 9%, and 78%, respectively. Kim et al. [37] analyzed the crystal structure of captured nanoparticles, which were generated in HFCVD. Various crystal structures of carbon allotropes, including cubic diamond, n-diamond, hexagonal diamond, and i-carbon, were identified. We analyzed that d-spacings of 2.06 Å, 2.2 Å, and 2.5 Å were related to various carbon allotropes, such as a cubic diamond, n-diamond, hexagonal diamond, and i-carbon. [37,38,39,40]. Vora et al. [40] analyzed the i-carbon film, which contained an unknown cubic phase of a lattice parameter of 4.25 Å. They confirmed that d-spacings of 2.43 Å and 2.12 Å of the phase were assigned to (111) and (200) planes, respectively. Shown in Figure 7c, the nanoparticle had the d-spacing of 2.5 Å with a polycrystalline structure. We observed that some of small nanoparticles had a single crystalline phase with d-spacings of 2.5 Å and 2.10 Å, which were assigned to (111) and (200) planes of a cubic phase with a lattice parameter of 4.25 Å. This phase was almost the same as i-carbon reported by Vora et al. [40]. The observed i-carbon nanoparticles had a variation of 2.36 Å–2.54 Å in the d-spacing value. As a result, there was a variation of 4.1 Å–4.4 Å in the lattice parameter. From this analysis, polycrystalline nanoparticles shown in Figure 7a,c mainly consisted of i-carbon.

Nanoparticles captured at 2100 °C were single crystalline with the size of 3 nm–5 nm. D-spacings of observed nanoparticles consisted of 2.06 Å, 2.2 Å, and 2.5 Å. These d-spacings were the same as observed for nanoparticles captured at 1900 °C. However, the observed relative fractions of 2.06 Å, 2.2 Å, and 2.5 Å were 26%, 32%, and 42%, respectively, which was quite different from those of nanoparticles captured at 1900 °C. It should be noted that the relative frequency of 2.5 Å d-spacing for i-carbon decreased from 78% to 42%., which means that nanoparticles captured at 2100 °C contained much less i-carbon than those captured at 1900 °C.

The larger relative frequency of i-carbon at the filament temperature of 1900 °C than that of 2100 °C can be attributed to a less amount of excess negative charges in the nanoparticles at 1900 °C than at 2100 °C. Lai and Barnard [41,42] examined the thermodynamic stability of hydrogenated nanodiamonds in both neutral and charged states. They showed that negative charging of hydrogenated nanodiamonds spontaneously desorbed atomic hydrogen from the surface. They also confirmed that anionic charging saturates dangling bonds of carbon atoms at the surface, resulting in the disappearance of reconstructed and graphitized layers at the surface. Park et al. [43] reported that negative charges stabilize the structure of nanodiamonds because the excess electrons saturate and stabilize the dangling bonds at the surface of diamond nanoparticles. If so, diamond nanoparticles generated at 1900 °C at 3% CH_4_ –97% H_2_, which are relatively deficient in excess charges than those generated at 2100 °C, would mostly have the structure of i-carbon.

The possibility that carbon nanoparticles generated at 1900 °C at 3% CH_4_ –97% H_2_ were deficient in excess charges is also supported by Figure 7c, which shows that diamond nanoparticles have a large polycrystalline structure. Such a structure would be formed by the agglomeration of nanoparticles, which would be induced by weak Coulomb repulsion between charged nanoparticles due to lack of excess charges. It should be noted that the diameter of nanoparticles in Figure 7c is three times larger than that of the nanoparticles captured at 2100 °C in Figure 7d. Furthermore, the increase in the size of the nanoparticles can also weaken the phase stability of the diamond phase, since the capillary pressure from the surface of the nanoparticle decreases in proportion to the increase of the diameter [44].

Therefore, a lower amount of negative charges at the filament temperature of 1900 °C than that of 2100 °C would be related to the generation of i-carbon nanoparticles in a higher frequency, resulting in the high etching rate and low growth rate shown in Figure 3. The Raman spectra in Appendix A reveal that the crystallinity of diamond particles in Figure 2 and Figure 3 was related to the number density of electrons in the gas phase. The ratio (I_D_/I_G_) of the diamond particle on the bare silicon substrate in Figure 2 and Figure 3 increased as the filament temperature increased from 1900 °C to 2100 °C. However, the ratio decreased as the methane concentration increased from 1% to 3%.

## 4. Conclusions

The deposition rate was much lower at 3% CH_4_ –97% H_2_ than at 1% CH_4_ –99% H_2_ at the filament temperature of 1900 °C in the diamond HFCVD process. This unusual dependence of the deposition rate on the methane concentration can be attributed to the higher content of the non-diamond phase, such as i-carbon in the ball-like diamond structure at 3% CH_4_ –97% H_2_. The higher content of the non-diamond phase appears to come from the lower amount of electric charges, which can, again, be attributed to the hot filament coated by the graphite at the filament temperature of 1900 °C and 3% CH_4_ –97% H_2_. The understanding that is made in this study as to the unusual dependence of the deposition rate provides a new insight of the growth mechanism of diamond particles. The contribution of carbon nanoparticles in the gas phase to the diamond growth on the substrate has been revealed more clearly since the study shows that the phase of carbon nanoparticles in the gas phase is related with the quality and growth rate of deposited diamond.

## Figures and Tables

**Figure 1 materials-14-00426-f001:**
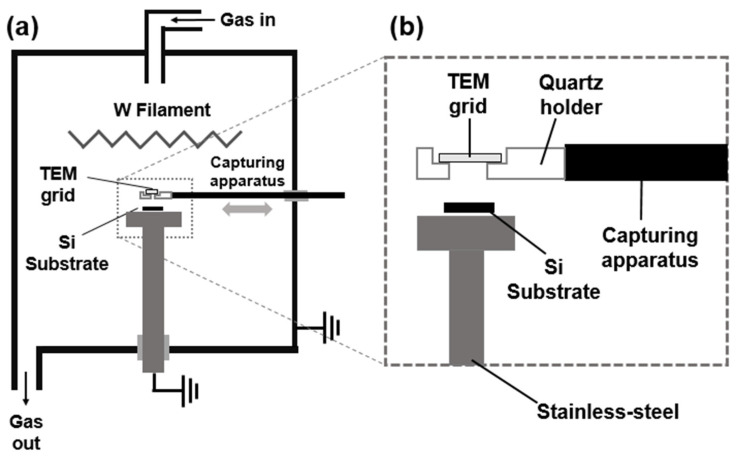
Schematics of (**a**) a hot filament chemical vapor deposition (HFCVD) reactor with the apparatus for capturing the nanoparticles on the SiO membrane of the transmission electron microscope (TEM) grid and (**b**) the capturing apparatus.

**Figure 2 materials-14-00426-f002:**
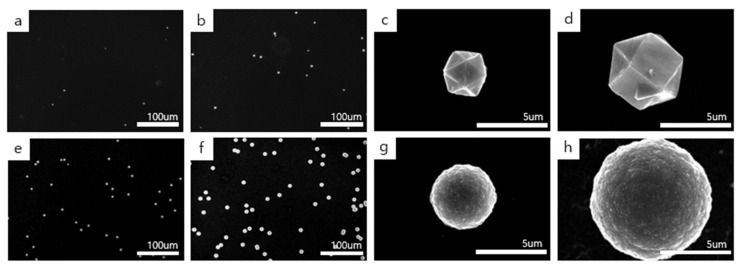
FE-SEM images of diamond particles deposited on the Si substrate at a filament temperature of 2100 °C. (**a**,**b**) show the low magnification image of diamond particles deposited for 4 h and 8 h, respectively, at 1% CH_4_ –99% H_2_, with (**c**,**d**) being the high magnification images, respectively, of (**a**,**b**). (**e**,**f**) show the low magnification image of diamond particles deposited for 4 h and 8 h, respectively, at 3% CH_4_ –97% H_2_, with (**g**,**h**) being the high magnification image, respectively, of (**e**,**f**).

**Figure 3 materials-14-00426-f003:**
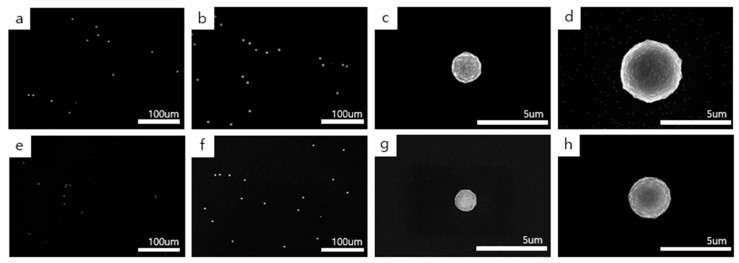
FE-SEM images of diamond particles deposited on the Si substrate at a filament temperature of 1900 °C. (**a**,**b**) show the low magnification images of diamond particles deposited for 4 h and 8 h, respectively, at 1% CH_4_ –99% H_2_, with (**c**,**d**) being the respective high magnification image. (**e**,**f**) show the low magnification image of diamond particles deposited for 4 h and 8 h, respectively, at 3% CH_4_ –97% H_2_, with (**g**,**h**) being the respective high magnification image.

**Figure 4 materials-14-00426-f004:**
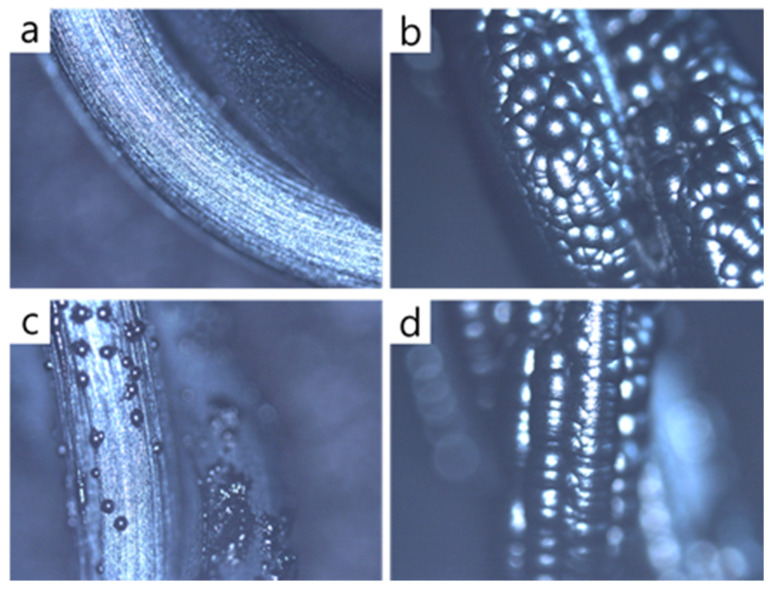
Optical microscope images of the filament surface (**a**) without the graphite coating at a filament temperature of 2100 °C at 1% CH_4_ –99% H_2_ and (**b**) fully coated with the graphite at a filament temperature of 2100 °C at 3% CH_4_ –97% H_2_, (**c**) partially coated with carbon precipitates at a filament temperature of 1900 °C at 1% CH_4_ –99% H_2_, and (**d**) fully coated with the graphite at a filament temperature of 1900 °C at 3% CH_4_ –97% H_2_.

**Figure 5 materials-14-00426-f005:**
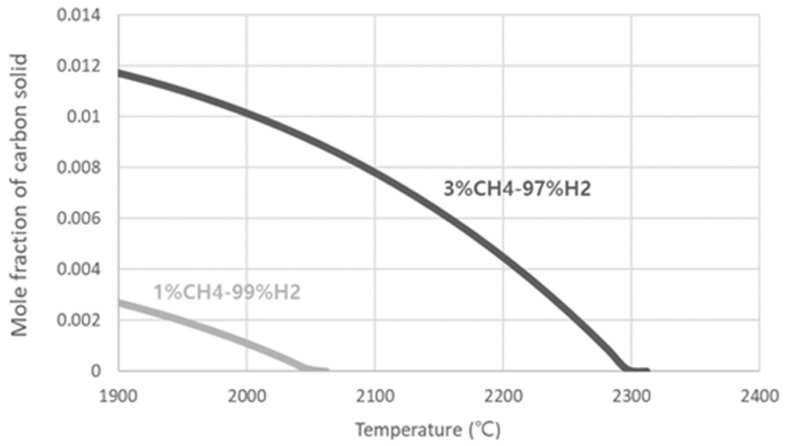
Thermodynamic calculations of the equilibrium mole fraction of carbon precipitation at methane concentrations of 1% CH_4_ –99% H_2_ and 3% CH_4_ –97% H_2_ by Thermo-Calc software using a Scientifica Group Thermodata Europe (SGTE) database [23].

**Figure 6 materials-14-00426-f006:**
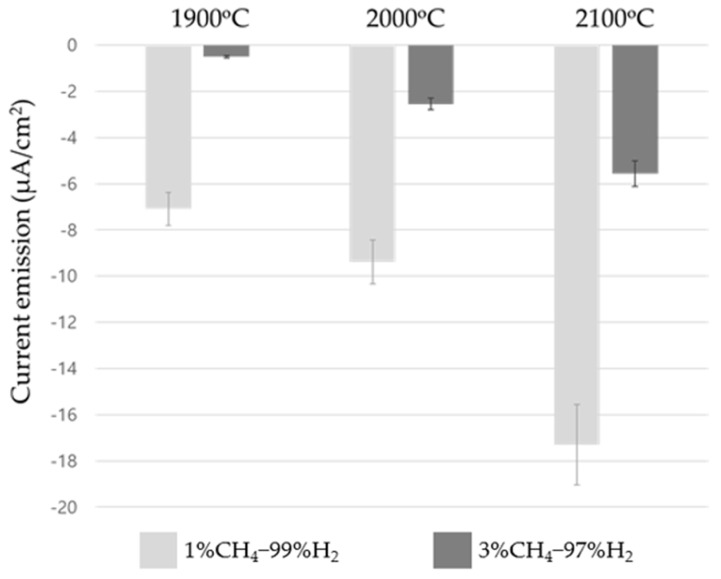
Current measured at filament temperatures of 1900 °C, 2000 °C, and 2100 °C at 1% CH_4_ –99% H_2_ (light gray bars) and 3% CH_4_ –97% H_2_ (dark gray bars).

**Figure 7 materials-14-00426-f007:**
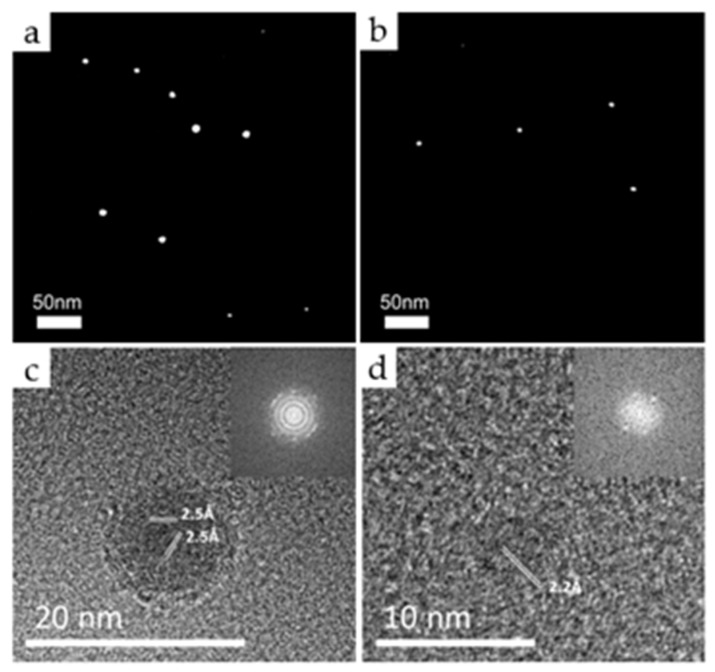
Transmission electron microscope (TEM) images of carbon nanoparticles captured for 15 s on the SiO membrane of the TEM grid. (**a**,**b**) show the scanning TEM (STEM) images of nanoparticles captured at 1900 °C and 2100 °C, respectively, at 3% CH_4_ –97% H_2_. (**c**,**d**) show high resolution TEM (HRTEM) images of nanoparticles, respectively, of (**a**,**b**).

## Data Availability

Data sharing not applicable.

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
