# Peer review of "Unusual Dependence of the Diamond Growth Rate on the Methane Concentration in the Hot Filament Chemical Vapor Deposition Process"

_materials, 2021, doi:10.3390/ma14020426_

Round 1

Reviewer 1 Report

The manuscript entitled "Unusual dependence of the diamond growth rate on the methane concentration in the hot filament chemical vapour deposition process" by Song et al. described the fabrication of diamond nanoparticles with different shapes at different conditions. They observed that the number density of nanoparticles is lower at the filament temperature of 1900 °C and gave reasonable explanations. The manuscript can be accepted for publication after some minor errors have been corrected and some texts improved. 

(1) The high-resolution TEM images of nanoparticles shown in Figure 6c and 6d are mislabeled as 6a and 6b.

(2) The term "i-carbon" needs to be explained or defined for readers who are not familiar with this abbreviation. Please cite the most relevant references.

(3) Page 5: “The main role of the filament has been believed to generate atomic hydrogen,” should be improved or rephrased to avoid unnecessary misunderstanding.

The authors should consider adding a short passage to explain the growth mechanism of diamond in HFCVD with relevant references, such as “In a HFCVD process, the molecular hydrogen (H2) that passes through the hot filament is dissociated into atomic hydrogen (H), while CH4 undergoes pyrolysis reactions leading to the formation of a variety of radicals. The production of atomic hydrogen simultaneously with hydrocarbon pyrolysis leads to diamond deposition, while the formation of graphite is suppressed due to the reaction of the atomic hydrogen with non-diamond phase.”

Reviewer 2 Report

This manuscript reports on the decrease in diamond growth rate with increasing methane concentration in a hot filament chemical vapor deposition of diamond at a filament temperature of 1900 Celsius.

The authors have presented their data well and the manuscript is well organized. However, enthusiasm for this manuscript is reduced due to the following scientific concerns.

(1) The authors have not grown continuous diamond films on the silicon substrate. Therefore it is not appropriate to call the present study as measuring growth rate of diamond which is typically measured in microns/hour. The authors are merely looking at the nucleation rate of diamond on silicon substrate under different growth conditions. At best, this study is providing only qualitative data without mentioning nucleation density, i.e., no of diamond nuclei per square centimeters on silicon surface. Since the nucleation of diamond nuclei is inhomogeneous, It may be misleading to compare a specific area on the silicon surface to draw conclusions. It is more appropriate to compare the average number of nuclei per square centimeters.  

(2) It is very well established in the diamond growth literature that the lowering of diamond filament temperature will reduce the plasma activation significantly; lowering the concentration of atomic hydrogen and concentration of activated carbon species required for diamond growth. Therefore, it is not surprising that lowering the filament temperature leads to higher concentration of non-diamond carbon.

Reviewer 3 Report

In this paper by Song and co-workers studied how CVD parameters affect the growth rate of diamond from methane. The concept itself is interesting, but there is not enough data to support the claims. The authors are invited to incorporate the following suggestions, which may validate the reported findings. This is necessary to reach the level of discussion expected from a publication in Materials.

1) The introduction section is very brief and contains only a single reference. The description of state of the art is summarized only in six lines, which is not acceptable. Please provide a thorough and critical description of this field. Show how it related to your findings. Highlight the novelty factor of this contribution.

2) Schematic of the setup used for the synthesis would be appreciated.

3) "By adjusting the distance between the hot filament and substrate, the substrate temperature could be targeted at 900 °C." - how was it measured?

4) What was the SEM acceleration voltage?

5) It is a standard to provide Raman spectra when nanocarbon is synthesized or modified. Please supplement these measurements in the revised version of the manuscript to enable validation of the crystallinity of the obtained material and how it changes depending on the synthesis conditions. Currently, the EM micrographs do not give a convincing answer if indeed diamond was the result of the synthesis.

6) Is the process reproducible? Have the authors tried to synthesize material more than once using the same conditions and compared the results? The observed trends need justification.

7) No error bars in Fig. 5.

8) Conclusions section should also include the description of the impact of this contribution and some future outlook. Overall, in this place and discussion of results, this study should be put in perspective.

Round 2

Reviewer 3 Report

I carefully reviewed the work again (the highlighted version was very helpful).

I am happy to recommend the publication of this article in the present form as I am satisfied with most of the answers.